# Multi-Agent Systems for Resource Allocation and Scheduling in a Smart Grid

**DOI:** 10.3390/s22218099

**Published:** 2022-10-22

**Authors:** Sami Saeed Binyamin, Sami Ben Slama

**Affiliations:** 1The Applied College, King Abdelaziz University, Jeddah 21589, Saudi Arabia; 2Analysis and Processing of Electrical and Energy Systems Unit, Faculty of Sciences of Tunis El Manar, Tunis 2092, Tunisia

**Keywords:** multi-agent systems, smart grid, distributed systems, residential buildings, commercial buildings

## Abstract

Multi-Agent Systems (MAS) have been seen as an attractive area of research for civil engineering professionals to subdivide complex issues. Based on the assignment’s history, nearby agents, and objective, the agent intended to take the appropriate action to complete the task. MAS models complex systems, smart grids, and computer networks. MAS has problems with agent coordination, security, and work distribution despite its use. This paper reviews MAS definitions, attributes, applications, issues, and communications. For this reason, MASs have drawn interest from computer science and civil engineering experts to solve complex difficulties by subdividing them into smaller assignments. Agents have individual responsibilities. Each agent selects the best action based on its activity history, interactions with neighbors, and purpose. MAS uses the modeling of complex systems, smart grids, and computer networks. Despite their extensive use, MAS still confronts agent coordination, security, and work distribution challenges. This study examines MAS’s definitions, characteristics, applications, issues, communications, and evaluation, as well as the classification of MAS applications and difficulties, plus research references. This paper should be a helpful resource for MAS researchers and practitioners. MAS in controlling smart grids, including energy management, energy marketing, pricing, energy scheduling, reliability, network security, fault handling capability, agent-to-agent communication, SG-electrical cars, SG-building energy systems, and soft grids, have been examined. More than 100 MAS-based smart grid control publications have been reviewed, categorized, and compiled.

## 1. Introduction

Energy is essential for human health and behavior since it powers transportation infrastructure and daily amenities. Polluting and non-renewable fossil fuels are currently the principal energy source for worldwide use. Investments and participation in complementary, clean, and renewable sources, such as wind and solar systems, have grown in tandem with their environmental consequences [1]. However, present usage surpasses natural regeneration, resulting in resource depletion unless more resources or alternatives are utilized [2]. To address these problems, it is critical to understand clean, alternative, and renewable energy sources and energy efficiency and sustainability. Traditionally, the Supervisory Control and Data Acquisition (SCADA) system has been used to orchestrate and communicate high-power directed systems that require a significant amount of analytical power to make intelligent decisions for the entire system, placing a significant burden on the central processor [3,4]. Because of its capacity to divide the whole system into many distinct controllers, allowing each unit to monitor each component in real-time, distributed control was deemed versatile, dependable, secure, and cost-effective [5]. Because the distributed controls attempt to tackle the problem locally, an outage in one system component does not instantly affect the others in this configuration. Many benefits of distributed generation (DG) include better dependability, enhanced security, and lower fuel costs. Solar, thermal, wind turbines, small bioenergy, and fuel cell systems are examples of distributed energy resource (DER) systems that can be freestanding or hybrid systems with various DERs [6]. As a result, less information must be sent to higher levels. As a result, the old electric grid has been changed into a smart grid, becoming a highly dependable and dazzling infrastructure, paving the way for the notion of “Smart Grids (SGs)” [7].

SGs employ sophisticated sensing, superior control, and communication. In this vein, SGs combine digital and electrical technologies to analyze and transport data swiftly [8]. Based on services such as real-time flow management, network operators, network operation, and individual consumption regulation, digital technologies aim to target production, transmission, distribution, and consumption [9]. SGs integrate power systems, networking, and communication technologies to improve the electrical grid. More real-time monitoring of SGs is required to enhance energy flow and facilitate conversation between providers and customers. To decrease carbon emissions, SG enterprises are increasing their use of renewable energy sources. A traditional distribution system allows efficient energy generation and storage [10]. Electric vehicles are an excellent method to store energy while also reducing pollution. The first power strategy incorporates network and network layers [11]. While making local decisions, the I-Intelligent Energy Electronic Device (IED) may manage power flow and equipment functioning [12].

I-Energy assures fair rates and a dependable electricity supply based on cost-benefit analyses. As a result, a SG must obtain all of its investment capital from clients, which has earned it a lot of negative feedback [13]. Numerous concerns regarding the economy and security of the smart grid, as well as how to improve and safeguard it against external and internal attacks, have been raised. The I-Energy technique has converted SGs in our homes into intelligent installations. Smart metering improves energy monitoring and regulation [14].

Multi-agent systems (MAS) can independently control power system operations. MAS contains several files. Intelligent creatures work together to solve problems beyond their capabilities [15]. MAS has been used in power market modeling, network protection [16], troubleshooting, and network control in recent years. The IEEE Power Engineering MAS Working Group reviewed MAS technology approaches, standards, tools, concepts, methods, and challenges in 2007.

SG technology is changing due to technological advances, security concerns, regulations, and environmental concerns. SG technology reduces the need for central generators. Over the past two decades, the electrical industry has developed wholesale power markets based on the decentralized decisions of generation companies. In addition to government regulations, consumer demand for clean energy is driving it. Photovoltaics (PV), fuel cells, and wind power are all standard in today’s electrical grids. After the Northeast power outage of 2003, home power generation systems emerged as a significant competitor in supplementing existing power infrastructure to bridge the gap [16]. In the aftermath of Hurricane Sandy, several government agencies urged investment in vital energy resources.

The smart infrastructure allows bidirectional power and data transmission based on energy, intelligence, and connection. T and D lines are used in traditional one-way networks to transmit electricity to load locations [17].

A SG, on the other hand, allows clients to create their power. Bidirectional energy flow is enabled via small-scale power generation. Microgrid and utility grid tiny auxiliary generators might be conventional or unconventional. The intelligent communication subsystem tracks and measures SG production and consumption. Energy efficiency, demand profile, energy loss prevention, cost and pricing, optimization, machine learning, and control processes are all required for intelligent management systems [18]. These involve management as well as infrastructure. Intelligent security addresses the issues of dependability, prediction, translation, and security. These observations are required for monitoring and measuring. They have predefined time windows for calculating depreciation and billing criteria. The changed measured data is transferred to the management system through wired or wireless networks. AMI is an enhanced version of the traditional AMR and AMM systems. Smart meters, home area networks, and broad networks are among them [19]. The expertise of a SG includes dependability, forecasting, localization, and offshore security. Intelligent protection systems troubleshoot, diagnose, self-heal, and safeguard small networks. The SG’s dependability is dependent on the DG’s sustainability, as intermittent RES and load-related oscillations should be minimized. Measurement and control systems are critical to service dependability and quality. The architecture for Predict and Prevent also aids in defending the SG [20].

The system or operator should rapidly diagnose and remedy the fault. Intelligent protection is divided into two stages: pre-fault and post-fault. Continuous monitoring of voltage, current, temperature variations, steady state, and transient attributes is used to carry out the phase. This monitoring helps to avoid mistakes. Measurements are used to discover and diagnose errors. The communication network comprises project management units, AMI, AMR, and other sensor networks [21]. The authors in [22], mentioned the intelligence network, including DG sources, traditional generators such as CHP, fossil fuel-based power plants, RES, electric vehicles (EV), and smart and smart buildings. Indeed, they reported that the communication between homes and data centers and the SG system should provide quality service, dependability, coverage, sustainability, security, and privacy. In [23] the authors cited that the communication infrastructure quality determines transportation safety. On the other hand, a dependable communication system with diverse designs connects numerous nodes and systems.

In this article, we make the following contributions:

This study examines smart grids and communication networks, considering relevant technology, uses, and issues. The document describes the SG, smart grid communication technologies, smart grid security concerns, smart energy infrastructure, and smart metering. The current state of each system and probable future research directions are summarized in the subsections. Electricity generation, transmission, distribution, and client facilities are all included in the first section; SG and Smart Energy Infrastructure. Then, smart metering and measuring applications are assessed using energy management and control systems and reference standards. This article discusses smart metering systems’ communications and security hardware and software infrastructure. We used the King Abdelaziz University Database (KSA) to conduct this research and only used the literature published within the last five years. A list of acronyms used throughout the paper is presented in Table 1.

The rest of the paper is organized as follows: A comparison with related survey articles is presented in Section 1. In Section 2, we provide Preliminaries: Detailed Analysis of the Literature. In the next section, we highlight the motivation for employing Intelligent Agents in SGs. Moreover, case studies on the use of Multi-Agent System Classifications are also presented in this section. In the same section, we discuss the taxonomy of the Blockchain Smart Grid concept. Section 4, the progress and development of smart home agents in prediction algorithms are discussed and surveyed. Finally, Section 4 concludes the paper. Figure 1 depicts and explains the various sections of the paper.

## 2. Preliminaries: Detailed Analysis of the Literature

As additional distributed energy resources (DER) are added to the power grid, a decentralized system for scheduling and allocating resources in an intelligent grid becomes necessary. Economic Distribution (ED) and Unit Commitment (UC) are crucial factors when distributing network resources for stability. The uncertainty around renewable energy makes resource allocation more challenging for system operators. More renewable energy sources and electric vehicles with bidirectional energy flow will be integrated into the future grid. This complicated smart grid system requires decentralized resource allocation, inter-node communication, and decision-making. Multi-agent systems (MAS) can decentralize the smart grid’s central resource allocation [24,25]. The three essential components of an intelligent grid are depicted in Figure 2: (a) the utilization of smart meters, devices, and applications for usage monitoring and management, (b) distribution of power generation to expedite the use of renewable energy sources, and (c) real-time control of the transmission and distribution network.

### 2.1. Energy Demand Side Management

Demand Side Management controls the network’s demand side and targets end users. The authors posted in [26] that through continuous monitoring and appliance management, these technologies cut energy usage in households, businesses, and industries. The work of two residents in the building influences the electrical energy consumption. It includes energy-efficient building systems, demand-driven control, demand response, intelligent buildings with smart devices, and energy dashboards [27]. These methods help utility providers and consumers to control electricity usage swings. End users will be more active in network and building operations via demand-side management [28]. Demand response (DR) is a growing demand side management strategy that changes energy use patterns. When electricity supplies are limited, building users must reduce peak consumption [29]. This implies measures to encourage consumption. The demand response reduced the commercial building’s peak power usage by 15 MW to 13 MW in 15 min and 11 MW in two hours. Through a smart automated response to demand, a large office building can reduce peak load by 25% [30].

#### 2.1.1. DSM Architecture

DSM’s communication and market design contains five levels. It is a consumer colony where each residence has solar panels or wind turbines [31]. All customers have the primary intelligent grid management system, which includes load, car/battery, and supply agents. The source agent delivers all source data in cooperation with the EPA and other climate data [32]. DSM methods may be characterized based on timing and customer performance (Appendix A):(1)Energy Efficiency/Conservation Agent (EEA)(2)Time-of-Use Agent (TOUA)(3)Demand Response Agent (DRA) or Payload Transfer Agent (LSA)

EEA encourages energy-saving technology and improves building design and construction. This category includes end-user comprehension and behavioral shifts toward more energy-efficient equipment [33]. DRA can short- or long-term shift load to peak hours for LSA. DR supports variations in power use in reaction to SG events, even as time-of-use bills reorder energy at different rates.

#### 2.1.2. An Overview of Leveraging Edge Computing in Smart Grid

To provide sophisticated communication and monitoring systems, the Internet of Things (IoT) has created small objects for communication, accessibility, or access to the Internet. In this vein, the productivity of infrastructures in various fields will be improved thanks to intelligent embedded devices and intelligent decision-making capacities, in particular, intelligent networks [34]. Like other areas of the IoT, the smart grid has been chosen as a potential technology using a wide range of sensors and information sources that gather data with higher resolution. One of the serious IoT issues is the processing of a large volume of data [35]. To address this concern, Edge Computing seeks to manipulate data to the integrated devices in which data is processed on the edge of the IoT infrastructure. Figure 3 illustrates an advanced design that uses the smart home to process information from the smart grid. The current electricity grid presents several challenges: unexpected disruptions and power outages, the theft of imperceptible customers, fixed electricity prices, etc. Such challenges contribute to the cost of electricity and to the growing demand for fossil energy [36]. The Smart Grid provides the following concepts to solve the conventional network’s lack of efficiency and performance [37]:Distributed generation: Customers can generate electricity from renewables, including solar or wind, in a smart grid. Surplus energy can always be supplied in the micro-grid to the grid or to other customers.Micro-grid: Low voltage electrical grids interconnected under clusters are defined as the micro-grid. This will improve the efficiency of local distribution through self-generated and controlled structures. The micro-grid can be connected to the grid. Nevertheless, if a malfunction, failure, or other risks occur on the grid, it differs from it.Smart meters: The deployment of smart meters allows the exchange of information between consumers and companies in real-time. In fact, smart meters monitor and track the additional use of smart devices in an apartment building.Dynamic pricing: This was chosen as the best tool for regulating electricity demand during peak periods. Depending on the total required power consumption, Time to Use (ToU), and real-time demand statistics, the total demand strategy can be determined.Proximity to prosumers: Data analysis was used to address automation issues closer to consumers, increasing the decision-making cycle and enabling consumers to choose to collaborate with a Virtual Power Plant (VPP) of any size or level.

**Figure 3 sensors-22-08099-f003:**
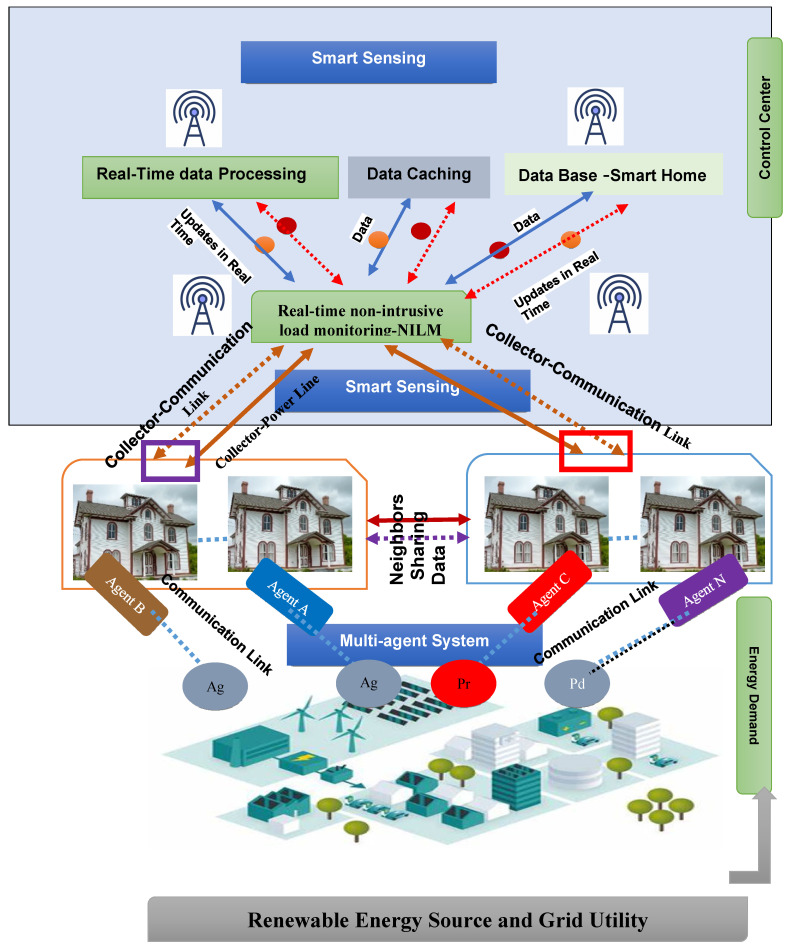
Agent in Smart Grid Model.

The smart grid is expected to provide a sustainable energy infrastructure through bi-directional data, energy flows through advanced knowledge, interaction, and communication infrastructure [38]. Prosumers are considered an attractive idea that will be an integral part of this smart grid. In this context, consumers can be recognized simultaneously as energy producers, energy sellers, and energy consumers. Prosumers not only consume energy but also participate in the network and/or excessive energy at the community level produced by renewable energy sources (such as solar, wind turbines, etc.) [39]. This seeks to solve environmental, social, and economic issues associated with high fossil fuel consumption. The smart grid encourages stakeholders to create communities based on various criteria, including energy usage, in order to manipulate consumers in the energy exchange system [40].

Consumers can choose from a wide range of electricity markets when using renewable electricity, and setting an average megawatt-hour for a specific calculation. Alternatively, power outages can be handled periodically to increase the battery capacity/storage system. The SG network contains several power connectors. To evaluate actual weather data from different regions, the prosumer runs the estimated energy profiles three times during the T period (given by (1)). Dependent on this period, virtual power plants and market models have been developed and have simulated actual production and consumption trends.
(1)$$\left\{ \begin{array}{l} \mathrm{Pr}os(\phi )=(g(P_{\phi }(T),type=\{P_{DEG},P_{EES},P_{FDA}\})\\ P_{\phi }(T)=(P_{\phi }(j)|j\in \{0\ldots T\},\phi \in \{1\ldots M_{\phi }\}\\ M_{\phi }=N_{FEA}+N_{EP}+N_{ESD} \end{array}\right.$$

Pr*os* (*ϕ*) represents the prosumer energy production; Pr*os* (*T*) is the estimated power; *P_DEG_* is the droplet-based electricity generator; *P_FDA_* is the provider energy, and *P_EES_* is the energy storage.

In order to provide standby power treatment facilities, the Virtual Power Station (VPSs) integrates and coordinates domestic power generation sources with flexible storage and assets with manageable loads [41]. The VPS is perhaps the most closely related to the many innovations in the field of renewable energy over the past few years. Combining technology from two separate energy sources, the VPS ensures a stable supply of electricity, regardless of environmental or demand conditions. Both the positive and negative aspects of the VPS are present during peak periods, as they can work well together. Thus, it allows better use of renewable energy technologies. However, the multi-source structure creates difficulty for companies to track, organize, and maintain the health of the system. It also requires better applications where different systems must be interconnected. This local collaboration tends to allow the best use of distributed energy systems by utilizing local resources and through the participation of consumers in energy management processes, regardless of size. The aim of creating a sophisticated ideal consumer concept is to define a group of current energy consumers [42]. The specific group tends to provide the highest clear performance targets for a particular form of service that the VPS needs to access and improve its income per energy user. The VPS’s alliances consist of the N-length chain, which means that the value 0 at location k is not part of the alliance; whereas the value 1 indicates that the consumer has been incorporated (taken) into the alliance. The VPS is expressed by (2) [43].
(2)$$\begin{array}{l} \left\{ \begin{array}{l} VPS=\{(taken_{\phi },\mathrm{Pr}os(\phi )|\phi \in \left[1,M_{\phi }\right],taken_{\phi }\in \left[0,1\right]\\ taken_{\phi }=\left\{ \begin{array}{l} 0,\mathrm{Pr}os(\phi ),Is\;not\;a\;part\;of\;VPSs\\ 1,\mathrm{Pr}os(\phi ),Is\;a\;part\;of\;VPSs \end{array}\right. \end{array}\right. \end{array}$$

To standardize energy management systems in the electrical grid and electric vehicles, an accurate practical method of communication has been developed. For future charging stations (Erse), the ISO/IEC 15118 standard and the IEC 61850 standard serve as interworking nodes between the wireless vehicle and the network control center [44]. Digital vehicle charging specifications, automated loading times, and dynamic billing models during and after charging are not only for charging but also for vehicle authentication (Value-added IEC 15118 services) [45]. IEC 61850 focuses on reliable network connections. Service providers and energy companies need to engage with a wide range of infrastructure users, from diverse energy sources to different end-user groups, including private homes, business parks, and electronic vehicle charging stations [46]. The connection is based on standard protocols to date. Intelligent control and automation technologies enhance energy consumption in electrical power networks. The SG also considers the use of unreliable power supplies. Moreover, due to the current uncertainty of energy production, some energy supplies face various challenges [47]. Modernizing the energy system also creates new factors. Therefore, control systems usually generate complex systems. A variety of strategies has been discussed to accomplish those targets. Whereas the recognition of agents is not private, the Multi-Agent Systems (MASs) is a group of autonomous entities that can evaluate and communicate. MASs will thus allow a realistic environment [48]. A SG adapts power systems and increases their electricity consumption. The use of MASs allows suppliers to operate individually to increase the device performance and reduce demand in centralized power management [49]. Figure 3 shows the importance of the real-time process for the system to connect all devices, sensors, and devices, and exchange all information. For all connections, it seems that the real-time mechanism is absolutely necessary for the network.

### 2.2. Blockchain in Smartgrid

#### 2.2.1. Background

Distributed energy systems adapt to a variety of threats, involving high electricity demand, increased cyberattacks, increasing renewable energy progress, and the migration to a smart home for massive IoT applications [50]. These challenges have caused tremendous pressure and require sustainable solutions and initiatives to achieve electrical system security and performance. Consequently, blockchain has been used in a variety of areas, such as the innovative and distributed smart grid [51], as a potential innovative technology that has gained a lot of attention. Blockchain infrastructure was also created from a unique data series.

The main feature of this blockchain technology is that it monitors all the changes in the chains it creates to avoid eliminating or replacing blocks [52]. This allows blockchain technology a very secure way to transfer resources, assets, and contracts without the need for a third-party authority, such as governments or institutions. Blockchain technology can guarantee stability, and facilitate connectivity, and payment security. In addition, blockchain guarantees the integrity of customer transaction data and the trust between stakeholders [53].

As a result, the blockchain approach has gained popularity, especially in the Bitcoin cryptocurrency sector. Indeed, blockchain smart grid technologies aimed to offer an innovative range and inexpensive approach to the tackled issues by existing and potential smart grids [54]. Numerous researchers have proposed certain technological solutions and concerns in the field of blockchain technologies for the SGs, which shows how the blockchain can be used as an SG cyber-physical layer [55]. This approach has encouraged energy innovators to assess their simple electrical systems without influencing the main energy system [56]. Many projects in the energy sector are approved to enforce a blockchain safety case by loosening small-scale energy policy to ensure innovation. The sharing of data between prosumers is, therefore, better than conventional methods of transmission [57]. Prosumers are viewed as both energy users and customers in the blockchain SG. A prosumer is independent of other prosumers, who do not generate energy for companies but produce electricity for their needs using a solar system chosen as an affordable and restricted usage approach [58]. Therefore, larger providers can examine the demand of small prosumers. According to the previous benefits, the blockchain allows remote settlers to generate and exchange their electricity in a common community with other stakeholders [59]. This allows the extra energy to be transferred to the grid. Customers in a specific neighborhood will benefit from energy by using public services such as mobile platforms. Rural families or isolated areas will produce their energy using renewable energy if they do not have access to electricity [60].

#### 2.2.2. Blockchain Technology in Home Energy Management

Managing energy demand and demand is becoming more and more challenging, especially as the need for renewable energy continues to increase. In this regard, DemandSide Management (DSM) should be used to coordinate market forces, enhance network stability, and expand existing network capabilities [61]. The current energy demand control project focuses on the interaction between both the system and customers. In the community of renewable energy production, centralized management of the demand side, and in the real-time context, however, the previous methodology does not depend on currency trading. There is a need for a demand-centric centralized control network for many households, with the addition of a smart meter network and sustainability [62].

Emerging blockchain technology may include privacy through the utilization of a digital network to enforce effective energy management approaches. It can execute a deceptive database that continually increases by maintaining a collection of data blocks deposited in a chain of the sequence. First seen in Bitcoins, this platform is often used as a bitcoin-monetary system [63]. With both the new algorithms and software, several activities may be performed automatically on the blockchain that communicate with data on the internet or even in the physical world. Effective procedures of smart nodes may be carried out on blockchains [64]. In a variety of scenarios, blockchain contributes to the Internet of Things (IoT), such as digital infrastructure, payment analysis, and social networks. So, these companies encouraged confidence, lowered prices, and improved safety. It is exciting to publish the Brooklyn microgrid network alongside other groundbreaking developments in chemistry and energy production from the blockchain. This gives it a somewhat prominent place in the concentrated power industry. Some major energy suppliers and start-up companies specifically seek to do this [65]. Most energy providers enable their representatives to purchase renewable energy shares as well as exchange an interest in ownership. Stakeholders will either sell or use the energy generated from their own property, and their assigned portfolios can be distributed simultaneously with their stake in the company [66]. For a variety of potential energy management investments, blockchain technology has many advantages in terms of business in the carbon market and in improving energy management and storage [67].

By separating the external stakeholders or mediators from the network, blockchain is used to reduce the cost of energy exchanges.Blockchain is being used to modify and monetize application transfers with distributed energy resources.Smart agreements were used to promote power exchanges at distributed energy resources level among customers.Blockchain is a shared database that preserves the time chains of data. Such blocks are not alterable or temperable.Blockchain facilitates multi-factor verification increasing distributed network blockchain.All activities carried out in a database that promotes transparency and accountability are accessible to the public.Blockchain offers distributed grid operations [68].

### 2.3. Smart Sensors Interoperability in SG

SGs are electric power grids that use advanced information, networking, and real-time monitoring and control technologies to save costs, conserve energy and enhance security, interoperability, and reliability. Smart Sensors (SSs) can provide real-time network monitoring, protection, and control information. Figure 4 illustrates the Smart Sensor model for GSs. In fact, the graph shows that SG networks suffer from interoperability and data interchange. According to the physical parameters, SSs transmit electrical pulses. There are options for digital or analog output signals. Data processing, analog-to-digital conversion, signal conditioning, and digital signal output are all features of digital sensors. Smart digital sensors could exist. Intelligent sensors and transducers are required for distribution network monitoring systems [69]. A communication format for power system phase measurements is described in IEEE C37.118. Data from sensors should have a reliable timestamp. Some of the standards for SG sensors are:Coordinated Universal Time Synchronization (UTC) and Exact Time. Fast data processing and intelligent algorithms (e.g., phase and frequency estimation, ROCOF from observed voltage, current, and time synchronization signals) [70].Precision and sensitivity detect voltage, current, and phase angle [71].Network and data transport with high speed, security, and dependability.Measurement accuracy and sensitivity to current, voltage, and phase angle [72].Fast, reliable, and secure network and data transmission.SS and Plug and Play interoperability require standardized testing methodologies and interfaces.Large bandwidth/dynamic sensors for the measurement of medium voltage (MV) of 600 V at 69 kV, of amps at kiloamperes, and 50 Hz at 5 MHz [73].Various sensing capabilities include voltage, current, power flow, temperature, weather, and climate.Sensors for self-identification, self-localization, self-diagnosis, and self-calibration are available.

**Figure 4 sensors-22-08099-f004:**
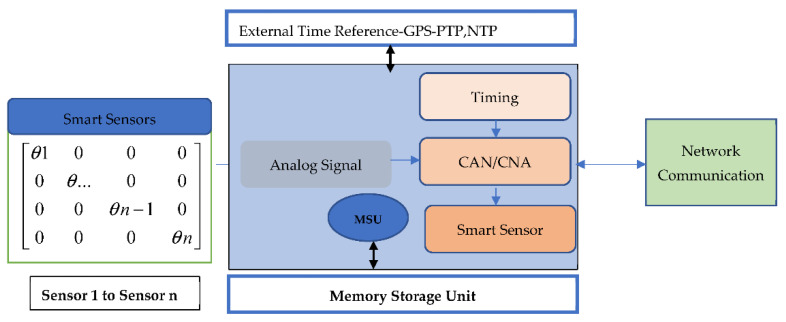
Smart Sensor model for SGs.

SSs are smart sensors placed in power grids, and sense temperature, pressure, humidity, weather stations, and current and voltage. SSs may communicate with the outside world using the IEEE 1451 Smart Power Adapter Interface standards, the IEEE 1815 Standard for Electrical Power Systems Communications—Distributed Network Protocol (DNP3), IEEE C37.238 PTP Power Profile, and others [73]. The following subsections describe the sensing, applications, and networking capabilities of the Phasor Measurement unit (PMU)/Measurement unit (MU)-based SS units. Table 2 compares and displays all Standards Specifications. There may be hundreds or thousands of SS units from various vendors in electric power systems. For electrical networks, SS interoperability is an issue. The first table provides a list of typical communication SSs protocols for SGs [74].

## 3. Intelligent Agent

### 3.1. Intelligent Agent Concepts

An entity (software or gadget) that operates in a particular environment and can freely interact with changes in that environment is referred to as an agent. Figure 5 compares and displays the agent classifications and parameter specifications. As shown, MAS employs task managers and communicators to acquire and regulate data. For example, great evidence was presented to show that sustainable power grid control using intelligent factors is locally determined to isolate the faulty part of the grid. The environment can be physical (like the power system), controlled by detectors, or the network environment (like data sources, computing resources, etc.), and it can be studied by making frame calls, engaging in communication, and receiving notifications [75]. The agent communicates with his surroundings physically or via recording diagnostic information for others in a database that is accessible [76]. IA mainly consist of four characteristics:Communication system;Decision making;Input and output interfaces.

**Figure 5 sensors-22-08099-f005:**
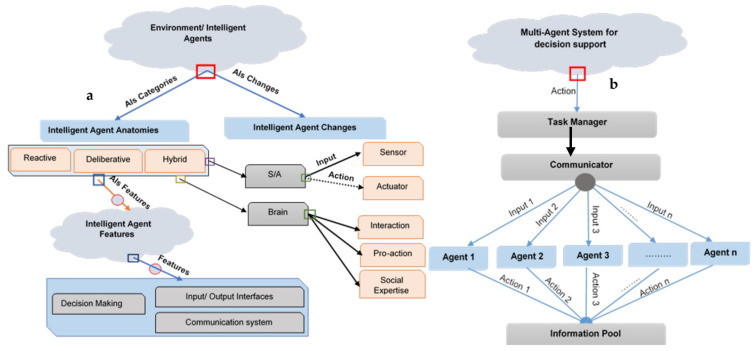
Intelligent Agent Model: (**a**) Agent Types; (**b**) Agent tasks.

The significance may theoretically apply to current technology and software. If the interpretation mentioned above is correct, the privacy sequence might be considered an “Agent”: the energy system is situated in its surroundings; it responds to environmental changes, such as voltage and/or current changes, and denotes a level of uniqueness (autonomy). In order to solve technological challenges, it is crucial to separate agents from Multi-Agent Systems (MASs) [77]. This requires expanding the definition of “Agent” to include “Intelligent Agent” (IA). The most precise definition of an agent is “an intelligent decision-making process that is positioned in some contexts and may work efficiently and independently for every transition” [63]. IAs were created by taking into account a variety of factors. The Foundation for Intelligent Physical Agents (FIPA) stresses that all data sharing between IAs occurs through the agent’s communication language [78]. Several agent-creation tools that adhere to FIPA criteria have been suggested and developed (see Table 3) [79]. In keeping with this, IAs exhibit flexibility and independence t comprising three key traits:(a)Interaction: Based on these modifications and the task of making them, IAs will reply appropriately and adopt a precise approach.(b)Pro-action: Putting IAs into practice fosters an analytical mindset that shows how the agent continually modifies his behaviors to attain his objectives. For instance, the agent may look for another agent who offers the same services if he loses touch with another agent whose services depend on him accomplishing his objectives [80].(c)Social aptitude: Intelligent agents can communicate with other intelligent entities. Social ability involves more than data transmission between conventional software and hardware. IAs have the capacity for cooperative and intelligent negotiation and interaction. The agent’s calling language often supports this capacity, allowing agents to converse and relay information [81]. To manage agents, many tools and platforms have been developed. Table 3 displays and discusses all agents’ fees. Agents’ frameworks are shown below:Java Agent Development Framework (JADE)Java-based Intelligent Agent Component ware (JIAC)Intelligent Agents SPRINGS (AgSpr)Intelligent Agents (JACK)Intelligent Agents Tracy (AgTra)Intelligent Agents (JADEX)Intelligent Agent Service (AgServ)

#### 3.1.1. Reactive Agents

These variables are mainly interactive and have no real understanding of their environment. It depends on the proposed first-use structure for Brooks. The layer hierarchy in this structure is represented by a specific behavior. Each layer or behavior is independent of the other layers in the system, but higher-level activity will remove the behavior at a low level [82]. This type of anatomy indicates the reasons for which it does not need to be proven, but it can be taken quickly if desired. The disadvantages of this form of anatomy are that the variables cannot change or overcome uncertainties and that the agent’s role in a particular situation may be difficult to anticipate [83].

#### 3.1.2. Deliberative Agents

The symbolic model of an illustrated scientist clearly contains the deliberative agents in which decisions are taken correctly. Agents of this type of anatomy develop plans to achieve their goals and therefore are more appropriate to address ambiguities and respond to unforeseen circumstances. Several models have been proposed with many BDI (Belief, desire, and intention) models to create commercial agents [84]. In this paradigm, what an agent feels towards himself and his world are his beliefs. The agent’s desires are his motivations, this is what the agent wants to achieve, but if the agent has several desires, then these desires collide. The agent’s intentions are the required action model to achieve the agent’s goals (the agent’s plans).

#### 3.1.3. Hybrid Agents

Throughout the action, most agents are a hybrid of reactivity and reflection, Turing Machines and Internap being examples of hybrid anatomies [85].

### 3.2. Multi-Agent System

A Multi-Agent System (MAS) is basically a system that contains two or more implementing IAs. There is no clear goal for a general system in a multi-agent system; instead, this is accomplished by bringing together several independent agents with each of them having a specific local target [86]. On the other hand, MAS has proven to be the most intelligent management unit that distributes the essence of power. The decentralized nature of the MAS method allows complex problems to be easily and cleverly divided into sub-problems by conducting simultaneous processing in order to meet its final goal, thus reducing the computation burden of a single device [87].

**Table 3 sensors-22-08099-t003:** Intelligent AGENTS’ Types.

Agent	Features	Agent Type	Ref.
Java Agent DEvelopment Framework (JADE)	It is a robust framework program. This optimizes multi-agent application deployment.	Agent Tools	[88]
Java-based Intelligent Agent Component ware (JIAC)	It is a structure of Java-based agents and a framework for developing and operating widely available software and services.	Agent Tools	[89]
Intelligent Agents SPRINGS (AgSpr)	The development of mobile, deployed and distributed computer applications is an extremely interesting technology.	Agent platforms	[90]
Intelligent Agents (JACK)	It consists of a set of independent factors that take inputs from the environment and communicate with other agents.	Agent Tools	[91]
Intelligent Agents Tracy (AgTra)	Used for mobile applications.	Agent platforms	[92]
Intelligent Agents (JADEX)	Jadex is the logical reasoning tool for Belief Desire Intention (BDI), which enables intelligent programming agents in XML and Java.	Agent Tools	[93]
Intelligent Agent Service (AgServ)	Prepare receipts, updates, records, applications or other documents regarding the obligations of another agent (customers/service providers).	Agent Tools	[94,95]

### 3.3. Smart Grid Agents

MAS was seen as a potential solution in SGs due to its ability to manage and secure operations in complex situations when operating as a centralized control system. MAS can efficiently identify disturbances, restoration of the electricity, control the secondary voltage and interpret the electric system [96]. Hierarchy would be preferable if the entire control system were to be transformed into MAS in which agents and subagents are considered separate objects. The SG agent model can essentially assess the impact of a hypothetical dynamic on the network [97]. SG agents can include two different classes of agents, one representing the problems related to the energy market, management of energy balance sheets, energy pricing, and energy planning [98]. In order to better understand how SG agents function, we should understand, firstly, the context of SG network management. SG Network Management (SGNM) and the correlation between SGNM levels is shown in Appendix B.

Second, we will concentrate on grid efficiency, security, and dependability problems. The functions of SG agents, such as hardware agents, Distributed Energy Resources (DER) agents, customer agents, intelligent control avoidance agents, smart response controls, and Graphical User Interface (GUI) agents, may be modified [99]. Typically, a multi-agent system serves as the coordinating and interactive layers. Based on predetermined information, the response layer initiates self-healing procedures right away. Based on the goals, the coordination layers confirm that the occurrence that prompts interactive action is more urgently needed [100]. A significant event can only reach the top layer if it goes above the predetermined limit. Information from the coordination layer is used to evaluate consistency. Between lower and upper levels, the MAS intermediate level maintains consistency with both agents. While the trading layer evaluates the system from a comprehensive viewpoint and allows planning for a longer time frame, the interactive layer has many agents that immediately impact the system’s behaviors and strategy.

In SGs, consumers are frequently referred to as “prosumers” since they consume energy and create power, and trade it with other consumers and producers [101]. Prosumer agents may combine the three critical functions of the seller, producer, and consumer. In this vein, SG Prosumer Agent is selected as one of the engaging ways to connect prosumer agents from various places. However, these prosumer groups, also known as Prosumer Community Groups, alter the operating system’s behavior, making balance and demand generation challenging to manage at all times [102]. Because of their comparable energy consumption patterns and interconnectedness, Figure 6 shows the SG-based control layers (PCGs) [103]. The interaction between distribution network agents (DSOAg) and transport system operator agents (TSOAg) is anticipated to change during the next several years, according to several upcoming technologies. Unpredictable generating units include centralized power and communications network infrastructure, which function partly as SGNM agents.

The SGNM can be managed better if we accomplish this using clever application agents. The network control and application management layers are the two functional levels that make up network management [104]. The voltage control in SGNM attempts to address various electrical network-related problems. In order to limit the quantity of network voltage to a specific value while considering other relevant concerns, factor-based systems must be deployed along with intelligent devices in place of traditional voltage management techniques [105]. It has been demonstrated that any modification to the grid impacts the entire system. The rising penetration of electric cars and heterogeneous generators into low-voltage networks also exacerbated voltage fluctuation. The network frequency is significantly impacted by changes in energy demand or power supply, whether there is a drop in demand or an increase in demand [106]. The frequency of presentations must be controlled at all times to provide a constant level within 24 h. Frequency fluctuations are lessened if a response to an agent-based request is included [107].

## 4. Application of Agents in SGs

### 4.1. Smart Home

Smart Home Agent (SHA) concepts are attractive because they serve residents reliably. Smart homes use innovative technology to enhance safety, healthcare, and energy consumption [108]. It is possible to remotely control and monitor home energy management systems (HEMS). HEMS include hardware and software to control the energy use of consumers (such as dealerships). Appliances offered home appliances ticking. The benefits of HEMS engineering in terms of safety and healthcare have been investigated [109]. Many prototypes improve energy use and improve the efficiency of energy systems. HEMS analyzes the energy demand in real-time using MAS devices. Household appliances and energy use can be scheduled to save electricity costs for the consumer. Home-to-home and car-to-home communications are new developments in smart home projects. During peak demand, the electric vehicle is used to power the home using vehicle-to-home energy storage technology [110]. Real-time smart home systems include sensing technologies, home network technologies, and devices. Due to design complexity and repetitive supervisor techniques without appropriate levels, the full potential of smart homes remains untapped [111]. PESM is characterized by some consumers who use and generate energy. Several studies indicate that central, and local governments can connect consumers to the network and to each other. Renewable energy is used in smart homes. For some of these customers, the typical energy loss is surplus energy at all times. The utilization of storage in the optimization system might be a future development.

#### 4.1.1. Prediction Algorithm in Smart Home

The smart house should include many Agent detectors to constantly perceive and govern the room. These detectors may be utilized to streamline typical processes, provide more services, and reduce outages. To deliver these services, smart homes should be able to forecast occurrences based on their ideas. Prediction algorithms are helpful for smart homes in this context.

#### 4.1.2. Markov Model

To understand how predictive algorithms work, first understand the context and ramifications. Understanding predictive theory and how a person communicates with the environment is challenging [112]. Some mathematical models are used in prediction algorithms to predict upcoming events [113] better. Algorithms simulate historical facts and theoretical results. User behaviors provide the predicted data for the smart home prediction system. These routines are called daily activities (ADLs) [114]. ADL is an activity that uses an agent’s sensor network to gather information [115]. Probabilistic models perform better than purely mathematical models because they require randomness, temporal uncertainty, and other factors [116]. Markov and Bayes’s models predict future outcomes based on previous iterations [117]. The Markov model believes that the system depends only on the previous direct state [118]. A random number can be represented as a different structure at a predetermined time to show how random variables differ over time.

#### 4.1.3. Bayesian Network Model

The BNM model is an algorithmic network model. Using a vector diagram, BNM displays random variables and their conditional dependency [118]. The structure and parameters are made up of BN blocks. Each edge represents a direct function between two BNM nodes [119]. Parameters are chosen using conditional probability tables or complicated learning algorithms. BNM analyses a device’s or signal’s behavior, gathers sensory data, and makes reasonable conclusions. They are categorized once enough data has been obtained [120]. The BNM model (P_BNM_) includes the state variable, z, which is considered as a fixed point, *ϕ*, which is determined by the gravity dynamics. BNM has several fixed values (*ϕ*i), each corresponding to the deep average range (μi). Finally, the (μi) model is chosen with just the confidence standard and expressed by (14).
(3)$$\left\{ \begin{array}{l} {}_{P_{{}_{\mathrm{BNM}}}=\frac{P(B|A)P(A)}{P(B)}}\\ p_{\mathrm{BNM}}{(z}_{t}=\phi_{i}{\;|X}_{1:t}>\lambda \end{array}\right.$$

*P*(*A*) and *P*(*B*) are Bayesian Network model probabilities.

#### 4.1.4. Prediction Algorithms in Smart Home Agents

SH agents have a four-layer structure, including the physical layer, which includes all home appliances, electrical connections, and network devices [121]. The communication layer maintains user agents and appliances internally and externally [122]. The specified and processed information layer offers knowledge for decision-making. The decision layer determines the operator’s actions based on prior information levels. Sensors monitor the environment and feed data to the following agent [123]. The database modifies and informs decisions by changing concepts and expectations in the data layer. They are executed in decreasing order (See Appendix C). The decision layer describes verbs using one or more prediction algorithms, assuming that generic behaviors are quite likely. Based on the prediction algorithm’s results, the decision layer can perform one or more actions [124]. The intended activity is automatable. Another alternative is to send alerts if the intended activity does not occur within a specific time [125]. Table 4 shows prediction algorithms used in innovative environments.

#### 4.1.5. Sequence Prediction via Enhanced Episode Discovery

The Sequence Prediction via Enhanced Episode Discovery (SPEED) algorithm was chosen as a potential algorithm to predict the interaction of inhabitants in a smart home [126]. The SPEED algorithm aims to recognize the natural trends of smart home users and try to make the right decisions based on the collected data. SPEED develops the Markov model with limited arrangements for describing the use PPM prediction algorithm. It is the speed with which residents communicate with their smart homes to predict future events inside the building [127]. Indeed, SPEED uses decision-making structures to construct a set of data, to which it can access knowledge subsequently collected to make a proper choice. SPEED immediately excludes any unnecessary information at any time during the meeting with the decision tree, which contains the majority of the k-level, in turn, classifying the 13th Markov Model. After a subsequent decrease in the full length of any specified sequence in the algorithm, the SPEED algorithm constantly computes the measured probability distribution [128].

#### 4.1.6. Flocking Algorithm

In a smart home, flow algorithms are useful for ADL analysis, especially when someone with Alzheimer’s benefits from them. Any specified collection of ADLs can function as a cluster, and a flocking algorithm can be used to effectively analyze large amounts of data [129]. There are other cumulative algorithms, such as K-mean, that can be used [130]. However, the K-mean design is not quite suitable for smart homes, as it needs individual approval and a preliminary combination [131]. This mostly includes the idea of using the flow algorithm as a common solution for a smart home because it does not allow any set of group numbers and separates the elements individually while giving the previously collected information. In addition, the flow algorithm can easily be adapted to the setting by introducing new connection points or by combining new data with suitable sections. One of the most impressive qualities of animal behavior in a robotics community is that decisions are made based on local knowledge, including sensory perception. However, as of today, most automated multi-agent systems focus on fully centralized position control or wireless communications, either from the performance capture system or from the Global Satellite Navigation System (GNSS) [132].

#### 4.1.7. Apriori Algorithm

The mining rules Apriori algorithm is a classic data mining algorithm. As shown in most research, a series of database standards are commonly used to determine repeated trends. Apriori Algorithm output are the rules of the setting for the “fXg!FYg” format, which includes “X” and “Y,” which is a section of all variable components [133]. No legitimate outputs are a rule in which both “X” and “Y” are empty. The law is defined as follows: in “X,” the frequency of the tests of the element’s “Y” [134]. This algorithm is generally used when studying transaction data in the store. High-Frequency (HF) differences can be found in this algorithm. HF patterns are directions for elements in a base that are usually referred to as lower-frequency support [135]. This algorithm is divided into several iterations, each repeating creating an HF chain. This is illustrated at every point of the data elements created or extracted from the market data.

#### 4.1.8. Nash H-Learning

Some learning algorithms can be useful in dealing with how smart homes adapt to their environment. This algorithm has been chosen to be attractive because it looks more efficient. Using this algorithm, a smart home can know the accuracy and rapidly of an employer’s habits [136]. A way to predict certain activities in a smart home is one of the learning algorithms seen as a potential solution. A significant amount of background knowledge should be given first to recognize how a Nash algorithm or learning system operates [137]. Nash H-Learning was created using the Q-Learning algorithm that deploys the Markov series in order to make the best decision because it is better embedded in a particular ecosystem. Hence, Markov is a well-established and useful process in the smart home environment even though Q-Learning still has its drawbacks and limitations [138]. Q-Learning is an advanced approach, but it is not appropriate for MASs structures because the environment is constantly changing due to external concerns.

#### 4.1.9. Active LeZi

The prediction has been chosen as an important solution to create smart systems with more intelligent and reliable decisions in a variety of areas within artificial intelligence and machine learning [139]. Many disciplines need to predict events that can usually be based on numerical methods and that are based on flows [140]. In this vein, Active LeZi is treated according to data theory. Active LeZi is a hierarchical knowledge theory predictor based on the well-known LZ78 family of compression data algorithms [141,142]. The efficiency of this algorithm is illustrated by using this algorithm to estimate system consumption in smart homes. The efficiency of this algorithm can be verified according to the data environments and standard experiences between the environment and the smart home. LZ79 has a decoder because it was designed as a compression algorithm. Coding is only required to approximate the basic sequence data of the remote test structure similar to the remote Markov model [143,144].

### 4.2. Microgrid Control

For distributed microgrid control, MAS can be developed in the microgrid [106]. As the MAS becomes automated, it is difficult to adequately control the individual DG by restricting the demand of the device operator; energy management must be restructured (see Figure 7). Indeed, MAS monitoring is a low-cost hierarchical control method [107]. Management, execution, and coordinating agents are the three levels of the MAS control system. Several stages lead to decision-making and observation. Sensors and field equipment are tracked from [145]. The coordination factor layer defines the execution points that satisfy the management layer’s policy. The management layer acts as protection against system limitations [146]. The Microgrid Monitor oversees and manages each of the following tasks based on three specific agents:(a)Management Agents: Efficiency, Economic, and Stability.(b)Coordination Agents: Generation, Negotiation, and Distribution.(c)Implementation Agents: Automation and Substation Operations.

**Figure 7 sensors-22-08099-f007:**
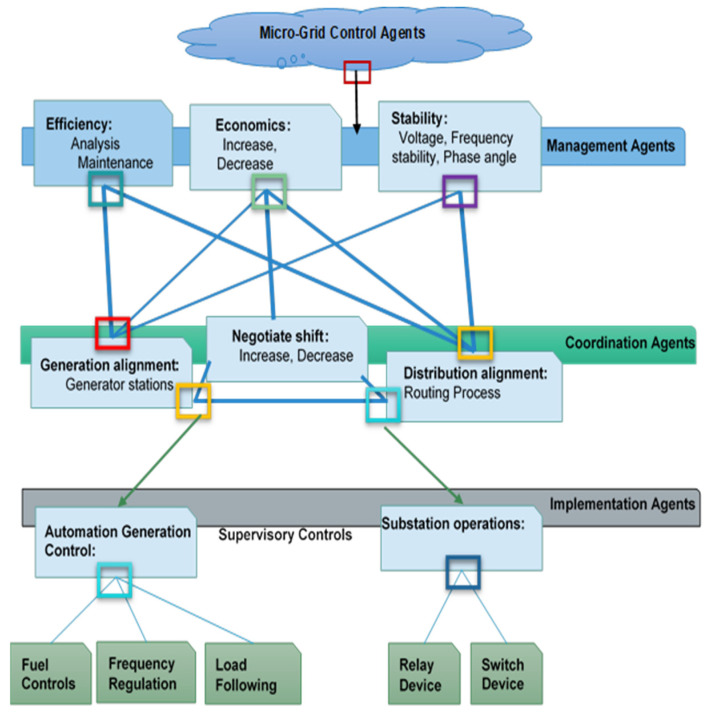
Microgrid Control Agents.

The centralized control layer is the microgrid control Centre (MGC). This later became known as the Policy Control Agent (PCA) in Microgrid. PCA centrally manages and controls GMs, ES and loads at three levels (See Figure 8):(a)Decision Control Agent—DCA(b)Management Control Agent—MCA(c)Scheduling Problem and Environment—SPE

**Figure 8 sensors-22-08099-f008:**
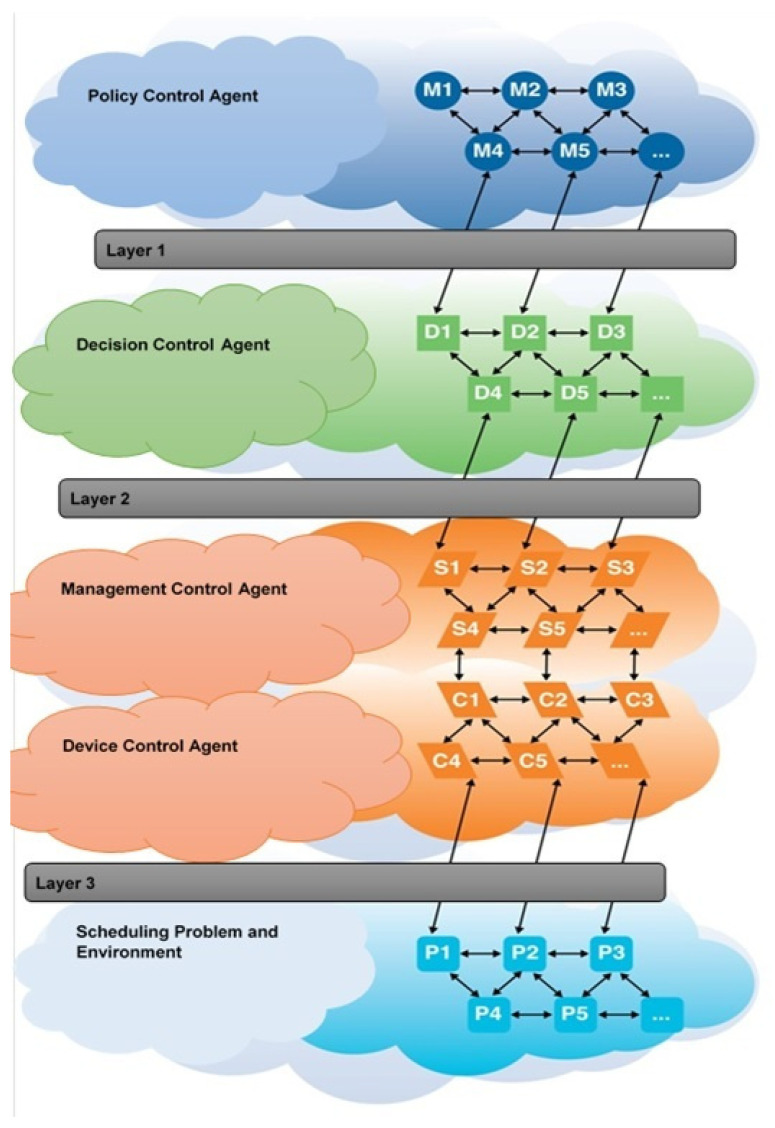
Policy Control Agent in Microgrid.

#### 4.2.1. Energy Balance Management Using AIs

SG’s energy management applications include Advanced Metering Infrastructure (AMI) and Home Energy Management System (HEMS) [147]. AMI is two-way communication technology, while HEMS is used to increase home energy in several ways. AMI and HEMS have been successfully used to control power consumption. Smart meter data reveals how, when and where consumer agents (PAg) use energy. Different approaches have been used to manage demand response (DR) programs, delivered at different times of the day based on facility requirements [148]. Demand Side Management is a DR Systems Implementation Technology (DSM). The DSM is an essential tool to meet energy demand and reduce energy expenditure for inverters of peak load and off-peak charging times. It adds to the balance of energy production. DSM perfectly implements smart grids as it prioritizes energy conservation and load balancing [149].

#### 4.2.2. Smart Meter Agents (SMA)

SMAs are chosen as possible devices for SGs. In this regard, SMA is an intelligent instrument that strives to monitor energy consumption intelligently and assists users in obtaining information about their energy usage [150]. SMA contains vital information that may be utilized to enhance energy efficiency and provides SGs with the capacity to enhance energy quality [151]. SMAs have been utilized for electrical load prediction, abnormality detection in electrical power systems, and other applications. Forecasting electrical loads offers a mathematical approach to power production and the construction of a system for managing production and demand for electrical power and is crucial to a cost-effective and secure operation [152]. The process of electrical load forecasting is segmented into long-term forward forecasting, medium-term load forecasting, and short-term forecasting. Annual, monthly, and hourly predictions are provided.

Prediction approaches have been utilized, including regression analysis, exponential smoothing, and weighted repetition [153]. Numerous complex techniques, such as neural networks and genetic analysis, have been employed to anticipate electrical demands. High-dimensional SMA presents several reliable pieces of evidence that are accumulating [154] (Appendix D).

It is challenging to predict electrical loads using SMAs directly. Some problems with assembly, such as high-dimensional data, were recorded. Significant data compression has been used as an essential solution to dimensionality reduction. Smart meters collect information about the electricity consumption used in real-time. Large amounts of data are sent over the network after phase analysis [155]. Companies can accurately determine the electrical load of customers. According to the analysis information step, it is possible.

### 4.3. Commercial Buildings

In most countries, the commercial building sector’s commercial office sub-sector generates most of the floor space and energy use [156]. The commercial office sector has grown steadily in most developed and emerging countries during the past decade [157]. In the past decade, the commercial office sector has been a target for improving energy management procedures [158]. Below, we describe MAS-based technologies to improve a building’s energy performance through a demand-driven response and control.

#### 4.3.1. Utility Agents

The authors reported in [159] how easy it is to install a MAS to coordinate heating systems in a workplace. One of the earliest practical applications of MAS has been as a utility agent, with choices based on the computational market concept/transient active control. In this case, cold air was provided to commercial representatives. Bidding agents bid for more or less air based on room temperature. The authors in [160] repeated the study and discovered improved results compared to the previous method. BECM systems must handle many occupant-friendly features in large commercial office buildings. According to [161], MAS has the potential to improve operating and energy efficiency. The authors constructed the MAS using utility factors and cross-decision methods. Active MAS has been used indirectly, through a decision-making process based on market value, to feed global information (building occupancy and comfort demand) into local technical parameters that change the behavior of individual systems over time. The system orchestrated the operation of chillers to reduce energy waste, particularly in partial or vacant rooms, based on on-demand global information based on building occupancy, which changes over time in large commercial office buildings.

#### 4.3.2. Simple-Reflex Agent

The authors in [162] developed a MAS framework based on inversion factors to govern office building systems as part of a more considerable investigation. The improved MAS system supports programmable agent limitations, which may be altered without having to rewrite the entire system. The MAS architecture is scalable and open, allowing context agents to be readily deployed and updated. Multi-agent systems may enhance building control and management as building function and tenant usage change over time. In addition, the authors [163] demonstrated that intelligent MAS control could save 20% more energy than traditional thermal management.

#### 4.3.3. Memory Agent

As intelligent agent systems are based on artificial intelligence, the solutions and concepts from this sector can increase building energy performance and comfort [164]. MAS provides learning and adaptive skills to govern building environments. The authors showed that the created approach increased user comfort by limiting control options. Using a MAS coordinate system with learning and memory states, the authors [158] showed that buildings might save energy. The authors in [165] found that adding IA approaches to an existing MAS-based building energy management system increased its performance. Using MAS can enhance the operating efficiency of a geothermal heat pump system and reduce gas usage by 23%.

### 4.4. Residential Buildings

According to the Residential Energy Consumption Survey (RECS), overall energy consumption in US households remained stable for many years as better energy efficiency has compensated for increases in the number and average size of dwelling units [166]. Indeed, residential renters are more price-sensitive to energy than commercial building occupants. MAS in residential buildings addresses the same challenge as MAS in commercial buildings, improving user comfort while minimizing energy costs, but takes a somewhat different approach owing to changes in the function of the building, type of device, and occupancy pattern.

#### 4.4.1. Utility Agent

With occupancy trends and charges, apartment buildings are a great target for meeting demand. Moreover, commercial renters must be rewarded so as not to lose interest. MAS systems in apartment complexes were reviewed to accommodate the demand [161]. Power matcher is a product of Dutch programmers [166]. Depending on the network throughput signal, the connected agents evaluated the energy efficiency. Ten CSPs were tested using analysis factors in the pilot study to demonstrate the system’s potential. The maximum load of the heating and electric units is half [160].

#### 4.4.2. Simple-Reflex-Agent

According to certain studies, using a basic inversion factor model and local energy management systems improves construction. In [162], the authors made similar observations to [160] while highlighting the benefits of MAS. According to the authors [163], centralized systems outperformed MAS systems using the matched information set. Scalability, openness, and adaptability were mentioned as advantages of one MAS over the other. Indeed, the authors discussed several applications of distributed optimization. They prove that the agent approach simplified regulation, coordination, and automation in innovative home construction. The authors [167] also demonstrated the ability of MAS to improve thermal comfort for building and management occupants, as well as its reusability and flexibility.

#### 4.4.3. Agents with Memory

Several authors have also demonstrated the use of IA concepts associated with MAS to provide residents with improved comfort and energy performance. Using the adaptive factor of the ability to learn the characteristics of the house, the authors demonstrated in [168] energy savings between 7.0–14.5% were achievable at home. The authors in [169] also demonstrated that agents with educational capabilities could, in addition to providing residents with improved comfort, improve the home’s energy performance by delivering wasted energy. In terms of improving the comfort index for building occupants, MAS has been shown to be able to continually identify and learn user patterns over time and make necessary adjustments to building controls without residents’ interference.

### 4.5. Energy Trading and Stock Exchange in Smart Grid

The electrical grids of the next generation will be SGs. They will enable homes, communities, and businesses to collect renewable energy and store it in a local battery, which can function as a microgeneration unit [170]. Customers may also be provided with dynamic electricity pricing, in which electricity prices fluctuate hourly in response to fluctuating energy grid demand. In a manner akin to the stock market, the end user can gain by exchanging energy with the grid. In [171], the authors investigated profit maximization for the end user equipped with renewable energy collecting devices and a battery, allowing the user to buy/sell energy from/to the grid by leveraging pricing and battery storage capacity. The program was executed without the knowledge of future energy needs, energy pricing, or renewable energy access—simulations based on tracking validated their findings. In [172], the authors validate the daily closing prices of 70 renewable energy stocks from 13 October 2010, to 4 March 2015. Pearson’s correlation coefficient is used to analyze PCC stock market communications. With simultaneous time series, the PCC determines the logarithmic exchange rate returns of stocks I and j and their similarities. To examine the topological properties of MST, three centrality scales are used: degree and centrality. Indeed, they concluded that First Solar Inc. And General Cable Corporation, and Trina Solar are more important within the network. These companies play a vital role in developing the market value of renewable energy. The authors [173] used random forests to predict the clean energy ETF stock values. Well-known technical indicators are among the features. Logit models predict stock prices less accurately than filling in a decision tree and random forest. For 20-day forecasts, random tree and forest packing yields an accuracy of 85% to 90%, while logarithm models yield 55% to 60%. Bagging trees and random forests are simple concepts to understand and estimate to predict the prices of clean energy stocks.

### 4.6. Non-Intrusive Load Monitoring

Non-Intrusive Load Monitoring (NILM) is a power demand monitoring and load determination system that employs voltage and current sensors at the power supply entry point. This approach minimizes sensor and installation costs compared to typical gas monitoring systems. An unbalanced three-wire 220 V/110 V physical single-phase home distribution system model was built and implemented in this study’s integrated Intel Atom system utilizing LabView software [174].

In [175], the authors stated that improvements in SM technology had generated massive volumes of data, enabling new opportunities for energy services and data-driven business models. NILM analyzes data collected from a single place to determine energy use at the device level. Deep learning is used in most modern approaches, resulting in models with millions of criteria. Some approaches calculate power consumption based on the target device’s brief work reaction, while others need the complete duty cycle. In the latter case, the split is either late (in minutes) or early (in seconds). This research suggests a real-time NILM system for rayon. The NILM algorithm identifies the target device’s functioning by assessing the observed active force’s transient response and calculating its consumption in real-time. The suggested system comprises three major components: event detection, a convolutional neural network classifier, and power estimation. The suggested system offers promising real-time results while being highly efficient in terms of processing and memory.

In [176], the authors stated that NILM is considered a potential topic of interest for universities and industry. By defining consumption by device or activity level, NILM can be used to unlock many intelligent home services and capabilities. The solution is deep learning. Most solutions focus on a selection of devices. In this vein, they have created a multi-class NILM system that can quickly identify any number of devices in this state. Transient feedback for active power measurements at 100 Hz is handled by hardware identification. The NILM system consists of adaptive threshold event detection, a convolutional neural network, and the nearest neighbor classifier. Future additions are selected automatically; no need for modeling or retraining.

### 4.7. Load Monitoring System and Applications

#### 4.7.1. Load Monitoring Concept

An instrument Load Monitoring (ALM) system can be designed in one of two ways, according to ILM and NILM. ILM is a traditional method for embedding sensors in each gadget and tracking consumption. Because of this, implementing the system is expensive. NILM needs one smart meter for each home. The smart meter’s data is divided for each device based on electrical properties and comes from a single source. ALM can be replaced by NILM, incorporating load identification and disaggregation techniques [177]. Energy shortage and climate change encourage energy efficiency laws and initiatives, particularly among consumers. Load Monitoring (LM) offers energy management solutions with real-time feedback. The deployment of traditional LM technology is costly and time-consuming. Indeed, LM is an Intrusive (ILM) and Non-Intrusive Load Monitoring (NILM) option [178]. As a result, to create an energy-efficient smart grid, smart homes must have Load Monitoring Systems (LMS). Software-based approaches and NILM are synonymous with hardware-based methods and ILM. ILM uses cheap home appliance gauges, but NILM requires only one sensor point. Although ILM solutions are more expensive than NILM, they are more effective and reliable. With sockets and other smart devices, future systems can incorporate NILM with a single energy sensor [179]. To design an efficient HEMS, identify and monitor primary household loads. Figure 9 shows two categories of operational management approaches: hardware and software.

The authors stated in [180], that the sensor measurements are used only in software-based (smart meter) methods. Because it requires only one detection site, NILM offers a low-cost approach. Most studies on this topic over the past five years have focused on these alternative solutions because, compared to hardware-based technologies, they have shown lower accuracy and higher complexity of implementation in real-world settings. To generate device profiles, NILM algorithms sample signals collected from smart meters using event detection. The collected signal can be rather noisy and only detect a few electrical devices depending on the sampling frequency.

The comparison between the two approaches is shown in Table 5. Unlike ILM, NILM uses only one sensor point (smart meter). Thus, it is not necessary to have a communication network that allows data exchange between the sensors and the home gateway. Although the reliability of NILM techniques remains an issue, these characteristics have enhanced their popularity.

In [181], the authors report that NILM demand control saves 20% of the energy. For this reason, several methodologies have investigated the basic steps of the NILM framework and provided a classification of device models with device signatures. Indeed, they highlight notable use cases and outline upcoming research issues, such as how to integrate intelligent meters and cloud computing in NILM to provide new seamless services for smart homes and creative grids. In [182], the authors stated that home automation and energy management is one of the best approaches to addressing sustainability challenges in the world. Common innovative home power grid scenarios have seen the use of NILM, including flexible load management, targeted energy efficiency improvements, and improved load modeling. They claimed that these technological advances might make intelligent home energy management systems easier to use. As a result, it may increase home energy efficiency and comfort levels while reducing energy waste and ensuring fair use of the home’s energy load.

In [183], the authors emphasized that energy monitoring is a critical component of energy management. Tracking a building’s energy use is required before creating technical solutions to reduce consumption. The technological case of device power management as it relates to ILM and NILM is covered in this study. ILM employs single-point sensing, while NILM uses distributed sensing. ILM and NILM have been used to analyze and rank various HEM systems to help academics in the region understand current energy management trends. The contributions of researchers and their methods in reducing the energy consumption of the building were also emphasized. According to the authors, more accurate identification and monitoring systems that detect as many pregnancies as possible are needed to monitor and treat pregnancy. More needs to be done by the NILM Energy Department. Energy users in homes, offices, and institutions need to be encouraged to manage their energy use. In [184], the authors note that LMS is necessary in light of current economic and environmental developments. Machine learning is used in this game to track energy usage, equipment performance, and even human activity. This article analyzes ILM research. Whereas NILM uses a single, smart meter as a single point of measurement, ILM uses low-end electrical meter devices built into the housing. ILM practices and ideas contrast with NILM. Additionally, features and machine learning methods are highlighted. In [185], the authors note that commercial building connection loads use a third of the energy. Researchers use smart plugs to record high-resolution consumption data to measure plug load usage. This data also helped create independent pregnancy detection algorithms to improve pregnancy monitoring. By reviewing the literature, they discovered several real-world implementation concerns, including limited publicly available datasets for commercial buildings, models trained on high-frequency data for sampling over a long period, and data leakage issues. They recommended calculating the plug-in load using low-frequency (1/60 Hz) powerful data. In fact, they showed that the dataset is processed by creating active component loading intervals before using dynamic time window feature extraction. These qualities are evaluated using precision scales. Multiple tests identify the best online and offline models, compare time window methods, and evaluate model performance under different sampling frequencies. The best online model achieved 93% accuracy, using a dynamic 5-min time window. Priority will be given to energy information panels and personal control systems.

#### 4.7.2. HVAC Systems

In an enclosed space, HVAC (Heating, Ventilation, and Air Conditioning) regulates the air’s temperature, humidity, and cleanliness. It offers good indoor air quality and thermal comfort—a mechanical engineering branch focusing on fluid mechanics, heat transfer, and thermodynamics. As in HACR (HACR-rated circuit breakers), “refrigeration” is occasionally added to HVAC&R or HVACR, or “ventilation” is dropped [186]. HVAC systems come in four different categories. Although each type of HVAC system operates slightly differently, they all ultimately contribute to maintaining clean air, specifically [187]:Heating And Cooling Split Systems—HCSS

The heating, ventilation, and air conditioning system includes heating and cooling systems. The system consists of a cooling unit and a heating unit. These are popular HVAC systems because they can be installed in most residences. Its installation requires no external needs or components. Installation may take a long time. Some companies recommend professional installation of partitioned systems.

Controlling Humidity Levels—CHL

Air purifiers and humidifiers with split systems are frequently used. This keeps your home warm no matter the weather. Dehumidifiers and humidifiers control indoor humidity levels. Users residing in dry or humid climates can benefit from these developments. Uncontrollable humidifiers and dehumidifiers are found in air conditioners and stoves. You can adjust the volume in your home using a different device.

Central Air Conditioner and Furnaces—CACF

The most common type of cooling system is central air conditioning. These units have indoor and outdoor components. Usually, these are split systems, but they can also be packaged systems.

Hybrid Split System—HSS

These are similar to heating and cooling systems. Their popularity is being increased due to their energy efficiency. These use ducts and thermostats, just like a split system. The alternative that saves energy is different—switching from conventional to hybrid heating systems. Gas heat pumps are noisier than electric ones. A propane, oil, or gas furnace uses gas power.

For example, in [188], the authors stated that HVAC accounts for 39.6% of the primary energy used in commercial buildings in the United States, representing 19% of total energy use. It is possible to save up to 42% on HVAC and energy costs by using wireless occupancy sensors or cameras for occupancy-based operation. Most of the solutions require building, implementing, testing, and maintaining sensors and their already existing network structures, which is costly. They report that Sentinel provides HVAC on an occupational basis using existing commercial building Wi-Fi infrastructure and residents’ mobile Wi-Fi-enabled smartphones. Sentinel is adaptable and works with existing building management systems. 86% of the time, Sentinel effectively detects workplace occupancy, with a false negative 6.2% rate. Powerful smartphone power management leads to inaccuracies. It was reported that Sentinel is used for powering 23% of a commercial building’s HVAC system for a day, and the electrical energy savings from the HVAC system is 17.8%.

In [189], the authors note that by monitoring and managing the load, conduction load management systems reduce the energy consumption of component loads in commercial buildings. Deplorability, power-saving capabilities, and system acceptance for real-world applications are all limiting factors. With an automatic innovative plug-loading process, Plug-Mate reduces load power consumption and human fatigue. The proposed system automates component loading based on users’ high-accuracy occupancy data obtained through a non-intrusive internal localization system, component load type data extracted through an advanced component load determination feature, and various control preferences through a customized user interface. Six control measures were tested throughout a five-month field study in a university office to demonstrate that the system was viable. To achieve the perfect balance between automation and human control, each control technology consists of different levels of component loading automation (manual, pre-scheduled, occupancy driven). Among the four component loads evaluated, the best control method demonstrated an average energy savings of 51.7%, resulting in a 7.5% reduction in building energy use and a 4.7 user satisfaction score. The feasibility of building-level implementation for upcoming real-world applications is highlighted in this work.

In [190], the authors mentioned that artificial lighting accounts for 19% of energy use in buildings. This requires energy-efficient lighting controls. *WinLight* is a revolutionary occupancy-driven lighting control system that seeks to reduce energy use while maintaining lighting comfort. *WinLight* calculates proper dimming instructions for each lamp using non-intrusive Wi-Fi occupancy data. A central lighting management system assigns these directions to the area gate, and occupancy-driven lighting is regulated by adjusting the brightness of each lamp. *WinLight* allows users to customize lighting and control nearby lights from their phones. Within 24 weeks, we implemented *WinLight* in a 1500 square meter multifunctional office in Singapore. *WinLight* saved 93% and 80% energy compared to a fixed-light control scheduling and a sensor-based lighting management system while ensuring occupant comfort.

## 5. Discussion and Perspectives

### 5.1. MAS Values

The new design must provide more advantages and value than the alternatives to be beneficial. The MAS system stands out for its scalability, manageability, and ease of implementation. For adding and deleting agents, MAS provides a flexible and reusable framework [27]. Because of MAS’s decentralized problem-solving features [28], smart buildings and varied network systems may be controlled, managed, and coordinated more easily [29,47]. As discussed in the articles, the MAS distributed nature and improved building operations management have been significant components of MAS that have raised energy performance and building comfort. This strategy is ineffective and unsustainable [48]. Recent debates have focused on successfully incorporating occupants into their comfort determination [49,79]. Providing occupants with enough adaptive skills for personal control may make them more tolerant and prefer more extended settings, hence improving energy consumption [83]. Giving construction users more control and engagement in day-to-day operations might be harmful if they make rash and ineffective decisions. This may lead to disagreements between users and administrators with competing goals (optimal convenience for users and optimum operating cost for managers). As proven by the authors in [80,82], MAS can manage these correlations to optimize construct performance (Table 6).

The scalability, modularity, universality, and ease of reconfiguration give MAS an advantage over competing systems, particularly when synchronizing user behavior and building activities. Advances in information and technology have made it easier to track user presence and behavior in buildings [86,89]. The matching of the energy demand for convenience by agents at the level of each building (workspace, room, building) was optimized using this information [88,91]. Energy: Reducing energy consumption in buildings is essential for two reasons:o It reduces expenses for building ownerso It promotes energy sustainability and reduces greenhouse gas emissions.

The distributed nature of the agent allows for the optimization of construction methods at any scale [92]. The balance between energy efficiency and occupant comfort in large commercial office buildings can only be achieved at the room level [93]. The interconnected structure of existing building control and management systems makes achieving balance challenging. Agent scalability, decentralization, and collaboration, as described in [98,131], facilitate the management of dependencies and interactions between systems, subsystems, and users, thus enhancing system efficiency and energy performance in buildings.

### 5.2. Setup and Operational Challenges

The objective of the MAS was to coordinate, organize and manage construction processes and interactions. Several kinds of research [28,160,161] have been conducted on the SG during the past two decades to confirm the ideas of MAS. However, it did not receive the wide adoption that its supporters had expected. Developers face several significant challenges in developing a compelling business case for MAS based on scalability, reusability, and ease of design [29,162]. Open data platforms, for example, require a DSM that includes interactions across multiple build systems and loads, which must operate as a single, interconnected system for optimal performance. The MAS coordination system must be aware of all these systems and be able to exchange information between agents associated with the systems and devices [163].

Open platforms are being used to assess April, Able, and Jade [164]. April is no longer maintained or updated, hence Jade is now the default platform. The practical deployment of multi-agent systems and the requirement for an intelligent enterprise-based information system are hampered by scope implementation issues [86,164]. DSM has issues as a result of combining random building occupants, building attributes, regulatory requirements and limits, power grid dynamics, operational demands, and market constraints. Due to cost, a lack of system support, mature technology, and logistical challenges, these systems stay at the theoretical framework level [59]. As a guide, only a few studies have been conducted involving complete interactions with power grids for load control based on multifactor systems (Table 6).

Hierarchical MAS systems for demand management entail assessing demand side resources to participate in demand reduction schemes. For demand management systems to succeed, supply and demand must communicate accurate information. Having many components from several suppliers and making time-sensitive decisions creates information-sharing challenges. Interconnection over multi-protocol gateways or devices necessitates independent configuration and, in some cases, vendor-specific tools [28]. First, multiple systems from many manufacturers need an open platform for information sharing. Many platforms are capable of supporting proxy-based infrastructure systems [164,165]. For proxy control and communication modeling, “Creating Intelligent Physical Agents—Agent Communication Language” and “Knowledge Query and Processing Language” are excellent protocols [166,167]. These languages have effectively enabled the use of ontology and collaboration between software agents in many sectors. Ontology is the general knowledge about a field familiar to many individuals and systems [16,31]. Using ontology to describe MAS design principles and what they signify provides a subtle semantic relationship between building system agents [100,104]. This provides the perfect amount of data from various sources in the construction process, for example, ontology, standardization, and data integration. The authors of most investigations recognized the language and semantics of inter-agent communication, which varied widely. A single ontology is required to unify the different building systems. The process’s massive volume of generated information is the second component of the information exchange load, which may prevent information exchange and decision-making [61]. Existing research lacks well-defined criteria for gathering sufficient information for demand side management systems, which require timely release and feedback. The agent framework aims to enable connectivity, independence, and mobility [28]. When choosing a platform for agent design, consider accessibility, modernity, manageability, and lightweight [28,31,100]. In order to enhance interoperability with other technologies and systems, agent design platforms must also comply with FIPA standards.

**Table 6 sensors-22-08099-t006:** Reported research and the MAS framework.

Ref.	Agents			Setup and Operational Challenges		Sector	
	Utility	Simple-Reflex-	With Memory	Real-Time	Virtual	Commercial	Residential
[27]	τ	τ	τ	τ	τ		
[29]	τ	τ	τ		τ	τ	
[98,131]	τ		τ			τ	
[98]	τ						τ
[28,160]	τ				τ		τ
[161]	τ	τ			τ		
[164]	τ	τ			τ		
[47]	τ	τ			τ		
[28]	τ				τ		
[61,68]				τ		τ	
[31,103]	τ			τ		τ	
[105,131]	τ			τ		τ	
[41,51,52,53,54,55,56,57,58,59,60,61,62,63,64,65,66,67,68,69,70,71,72,73,74,75,76,77,78,79,80,81,82,83,84,85,86,87,88,89,90,91,92,93,94,95,96,97,98,99,100,101,102,103,104,105,106,107,108,109,110,111,112,113,114,115,116,117,118,119,120,121,122,123,124,125,126,127,128,129,130,131,132,133,134,135,136,137,138,139,140,141,142,143,144,145,146,147,148,149,150,151,152,153,154,155,156,157,158,159,160,161,162,163,164,165,166,167,168,169,170,171,172,173,174,175,176,177,178,179,180,181,182,183,184,185,186,187,188,189,190]	τ	τ		τ		τ	
[2,39]	τ	τ		τ		τ	
[40,65,66,67,68,69,70,71,72,73,74,75,76,77,78,79,80,81,82,83,84,85,86,87,88,89,90,91,92,93,94,95,96,97,98,99,100,101,102,103,104,105,106,107,108,109,110,111,112,113,114,115,116,117,118,119,120,121,122,123,124,125,126,127,128,129,130,131,132,133,134,135,136,137,138,139,140,141,142,143,144,145,146,147,148,149,150,151,152,153,154,155,156,157,158,159,160,161,162,163,164,165,166,167,168,169,170,171,172,173,174,175,176,177,178,179,180,181,182,183,184,185,186,187,188,189]	τ					τ	τ
[40,69]			τ			τ	τ
[40,120]			τ			τ	τ
[99,123]						τ	τ
[107,135]							τ
[16,91,92,93,94,95,96,97,98,99,100,101,102,103,104,105,106,107,108,109,110,111,112,113,114,115,116,117,118,119,120,121,122,123,124,125,126,127,128,129,130,131,132,133,134,135,136,137,138,139,140,141,142,143,144,145,146,147,148,149,150,151,152,153,154,155,156,157,158,159,160,161,162,163,164,165,166,167,168,169,170,171,172,173,174,175,176,177,178,179,180,181,182,183,184,185,186,187]		τ					
[50,71,72,73,74,75,76,77,78,79,80,81,82,83,84,85,86,87,88,89,90,91,92,93,94,95,96,97,98,99,100,101,102,103,104,105,106,107,108,109,110,111,112,113,114,115,116,117,118,119,120,121,122,123,124,125,126,127,128,129,130,131,132,133,134,135,136,137,138,139,140,141,142,143,144,145,146,147,148,149,150,151,152,153,154,155,156,157,158,159,160,161,162,163,164,165,166,167,168,169,170,171,172,173,174,175,176,177,178,179,180,181,182,183,184,185,186]	τ	τ	τ				
[66,67,68,69,70,71,72,73,74,75,76,77,78,79,80,81,82,83,84,85,86,87,88,89,90,91,92,93,94,95,96,97,98,99,100,101,102,103,104,105,106,107,108,109,110,111,112,113,114,115,116,117,118,119,120,121,122,123,124,125,126,127,128,129,130,131,132,133,134,135,136,137,138,139,140,141,142,143,144,145,146,147,148,149,150,151,152,153,154,155,156,157,158,159,160,161,162,163,164,165,166,167,168,169,170,171,172,173,174,175,176,177,178,179,180,181,182,183,184,185]	τ	τ	τ				τ
[67,68,69,70,71,72,73,74,75,76,77,78,79,80,81,82,83,84,85,86,87,88,89,90,91,92,93,94,95,96,97,98,99,100,101,102,103,104,105,106,107,108,109,110,111,112,113,114,115,116,117,118,119,120,121,122,123,124,125,126,127,128,129,130,131,132,133,134,135,136,137,138,139,140,141,142,143,144,145,146,147,148,149,150,151,152,153,154,155,156,157,158,159,160,161,162,163,164,165,166,167,168,169,170,171,172,173,174,175,176,177,178,179,180,181,182,183,184]	τ		τ	τ			τ
[107,108,109,110,111,112,113,114,115,116,117,118,119,120,121,122,123,124,125,126,127,128,129,130,131,132,133,134,135,136,137,138,139,140,141,142,143,144,145,146,147,148,149,150,151,152,153,154,155,156,157,158,159,160,161,162,163,164,165,166,167,168,169,170,171,172,173,174,175,176,177,178,179,180]			τ	τ	τ		
[116,117,118,119,120,121,122,123,124,125,126,127,128,129,130,131,132,133,134,135,136,137,138,139,140,141,142,143,144,145,146,147,148,149,150,151,152,153,154,155,156,157,158,159,160,161,162,163,164,165,166,167,168,169,170,171,172,173,174,175,176,177,178]	τ	τ		τ	τ		
[19,108]	τ	τ	τ	τ	τ		τ
[13,43]	τ	τ	τ	τ	τ		τ

τ: This topic has been cited in this/these reference(s).

## 6. Conclusions and Future Works

This study focuses on the smart grid concept in consumer services, active distribution networks, and most smart potential network applications. In this vein, the SG was chosen as the potential power distribution system with major components. In fact, SGs were chosen as integrated information technology to manage and maintain electrical network data widely. SGs aim to develop real-time communications infrastructure to meet the growing needs of the power grid, including consumer bills, power load management, and network management across the entire network. It is also considered an advanced information technology due to its ability to handle large amounts of smart grid data.

Following identifying the promising potential of coordinated MAS systems, more work should be carried out to solve their weaknesses, such as defining a common ontology to promote interoperability and integration of MAS designs with other current design approaches. According to the reviewed papers, demand side management has many applications. Open platforms and complicated information exchanges hamper the adoption of multifactor systems. Excellent information collection should encompass all levels of abstraction creation to decrease the problems of diverse information interchange. The MAS concept may be divided into two categories: (a) extending classical multi-agent planning, and (b) economic model-based research. Whether you like it or not, the class employs classical optimization. Agents work together to achieve common aims. The second group based its selections on MAS market value. Both ways are valid. It is necessary and should be utilized to improve study findings. This is a finished design.

The improvements of V2H and V2G vehicle technologies will be included in our operations in the future to increase their coverage. This relates to developing an underground network and model for domestic electric cars. To do this, a challenging algorithm will be created and tested. Modern technology addresses energy consumption and cost concerns immediately. The focus of our effort in the upcoming phase will be on the creation and placement of the entire IoT platform in a lab setting.

The first notable restriction (Limitation) of our study is the quantity and nature of the source databases, even though the selected collections are generally reliable and representative. Second, the timely nature of the survey is constrained by the rapid evolution of this field. Thirdly, a synopsis of research efforts on various Multi-Agent System (MAS)-based intelligent home applications may not fully represent how the applications are utilized or influence individuals. Based on its findings, this review seeks to establish how the research community has responded to recent events.

## Figures and Tables

**Figure 1 sensors-22-08099-f001:**
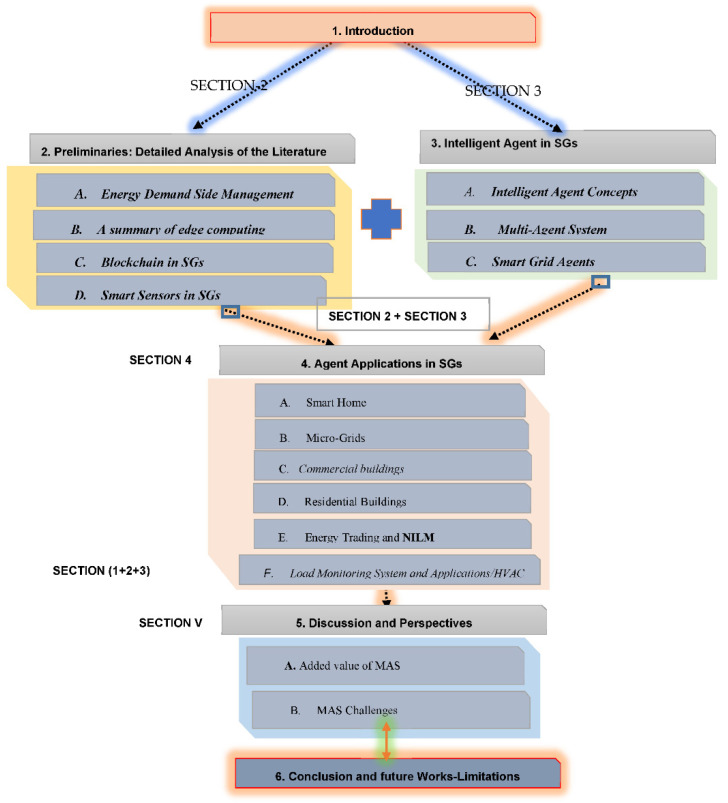
Description of survey paper sections.

**Figure 2 sensors-22-08099-f002:**
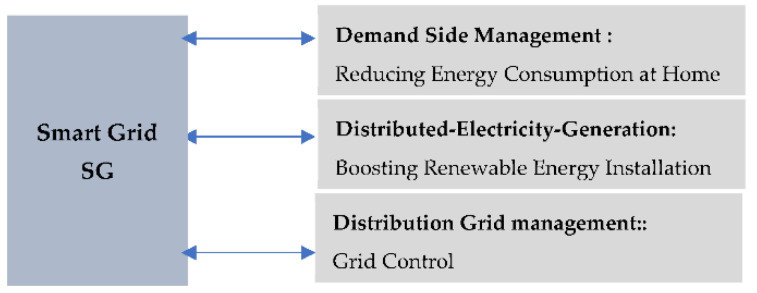
SG Components.

**Figure 6 sensors-22-08099-f006:**
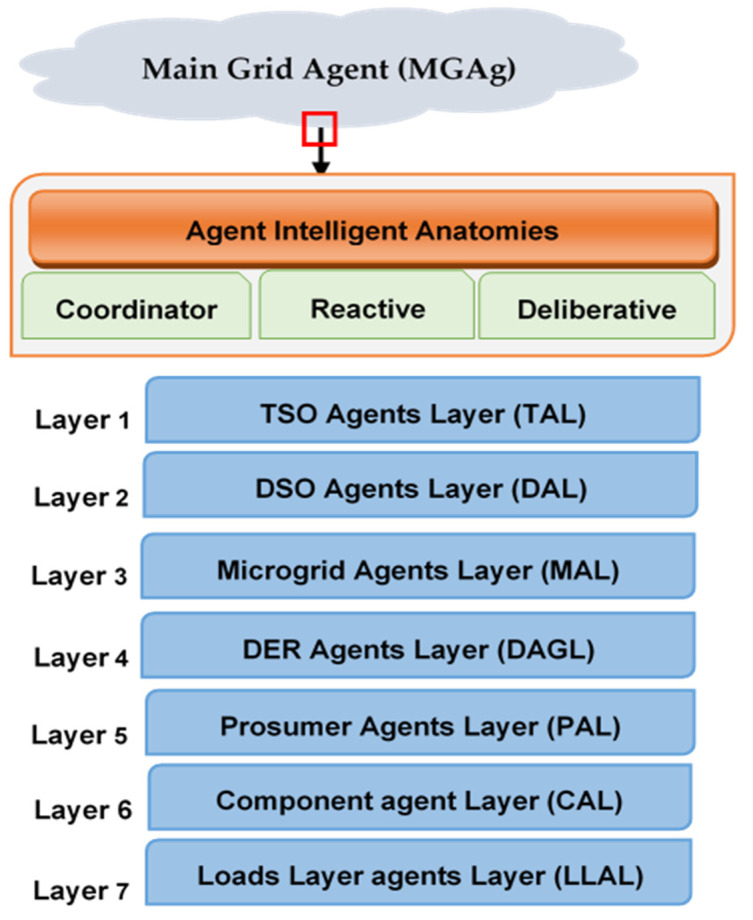
SG agents-based control architecture.

**Figure 9 sensors-22-08099-f009:**
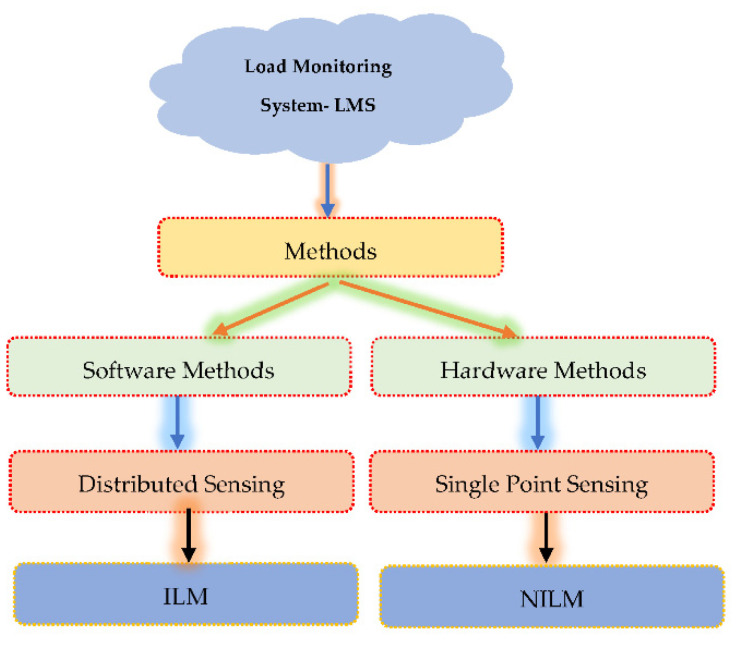
Load Monitoring Methods.

**Table 1 sensors-22-08099-t001:** List of acronyms.

Scada	Supervisory Control and Data Acquisition	DG	Distributed Generation
SGs	Smart Grids	DER	Distributed Energy Resource
SSs	Smart Sensors	NILM	Non-Intrusive Load Monitoring
IED	Smart Electronic Device	HEMS	Home Energy Management Systems
IoT	Internet of Things	I-Energy	Integrated-Energy
ToU	Time to Use	ACP	Active Consumer Participation
VPP	Virtual Power Plant	Pros (ϕ)	Prosumer Energy
VPSs	Virtual Power Stations	Erse	Charging Stations
IoT	Internet of Things	HEMS	Home Energy Management System
WCT	Wireless-Communication-Technologies	AWT	Average Waiting Time
PAN	Personal-Area-Network	UC	User Comfort
WAN	Wide-Area-Network	AMI	Advanced Metering Infrastructure
HAN	Home Area Network	AMR	Automatic Meter Reading
UCS	Uniform-Communication-Standard	GUI	Graphical User Interface
DNP	Distributed Network Protocol	SMDMS	SM Data Management Systems
PMU	Phasor Measurement unit	PSP	Power Scheduling Problem
PU	Measurement unit	ROM	Reliable Optimization Method
SMCS	Smart Meter Compression System	AOM	Approximate Optimization Method
DSM	Demand-side Management	ROAs	Reliable Optimization algorithms
DR	Demand Response	ILP	Integer-Linear-Programming
SSM	Supply Side Management	MILP	Mixed Integer Linear Programming
SGSH	SGs-Self-Healing	BFOA	Bacterial Foraging Optimization Algorithm
SGSM	SG Self-Monitoring	PSO	Particle Swarm Optimization
AP	Access Point	ACO	Ant Colony Optimization
NIST	National Institute of Standards and Technology	WDO	Wind Driven Optimization
SGTD	Smart Grid Transmission Domain	GA	Genetic Algorithm
SGMD	Smart Grid Markets Domain	TSA	Tabu Search Algorithm
SGUD	Smart Grid Utility Domain	KPMM	Knowledge Project Management Manual
SGDD	Smart Grid Distribution Domain	RAMM	Risk Analysis and Management Manual
ALM	Instrument Load Monitoring	ILM	Intrusive Load monitoring
HSS	*Hybrid Split System*	LM	Load Monitoring
CHL	Controlling Humidity Levels	CACF	Central Air Conditioner and Furnaces
HCSS	Heating And Cooling Split Systems-	NILM	Non-Intrusive Load Monitoring
HVAC	heating, Ventilation, and Air Conditioning	LMS	load monitoring systems
SGBGD	Smart Grid Bulk Generation Domain	MRM/IT	Manual on Risk Management for IT systems
SGCD	Smart Grid Consumer Domain	IEEE/EIA	Institute of Electrical and Electronics Engineers and the Electronic Industries Association
SGOD	Smart Grid Operation Domain	NIST	National Institute of Standards and Technology
PLCT/ICT	Power Line: Communication and Technology Information and Communication Technology	SOA	Service-Oriented Architecture

**Table 2 sensors-22-08099-t002:** SSs Standard Specifications.

Smart Sensors	Norms/Standards	Wired	Wireless
Basic SSs	IEEE 1815 IEEE 1815.1 IEEE 1451	TCP/IP UDP/IP RS232 Optical	5G LTE Cellular WiFi- ZigBee WiMAX 6LowPAN
PMU	-IEEE 1344- -IEEEC37.118.2- -IEC 61850–90-5-	TCP/IP UDP/IP RS232 Optical	3G/4G/LTE Cellular WiFi WiMAX
MU	IEC 60044–8 IEC 61869–9 IEC 61850–9-2	TCP/IP UDP/IP RS232 Optical	3G/4G/LTE Cellular WiFi

**Table 4 sensors-22-08099-t004:** Prediction Algorithm Categories.

Prediction Algorithms	Categories	Structure	Model
SPEED	Episode Discovery	Tree	Markov
Flocking	Clustering	Cluster	Rules
Apriori	Artificial Intel AI	Matrix	Rules
H-learning	Q-learning	User	Markov
Active Lezi	LeZi	Tree	Markov

**Table 5 sensors-22-08099-t005:** NILM vs. ILM.

Concepts and Features	ILM Method	NILM Method
Gathering Sensing points	Several sensors -Distributed Sensing	Single -Sensing-Point
Tremendous deployment	Hard deployment	Easy-Peasy to install
Trustworthiness and integrity	High	Less than ILM
Communications	Yes	No

## Data Availability

Not applicable.

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
