# Peer review of "Multi-Agent Systems for Resource Allocation and Scheduling in a Smart Grid"

_sensors, 2022, doi:10.3390/s22218099_

Round 1

Reviewer 1 Report

MDPI Sensors Journal (Manuscript ID: sensors-1943862)

Comments to the Author

This paper provides a review of the latest studies on smart home energy management systems as well as provide a comprehensive overview. It is an important topic and the paper studies the concept clearly. However, there are several points need to be addressed to improve the quality of the manuscript.

Suggestions to improve the quality of the paper are provided below:

1)     The authors should provide more information about how the review is conducted to demonstrate its comprehensiveness. What are the search terms used? Which search engines or database were used in this review? What is the search criteria used during review? What is the search period (i.e., limited to literature published within the last 10 years) of this review? Etc.

2)     The authors can also provide some high-level quantitative analysis of the papers reviewed such as the number of papers published over time, an analysis of the keywords used, the changing popularity of each topic over time, etc.

3)     Under Section 4 Home Management systems, the authors should also focus on load monitoring technologies and their applications. Please review the following papers as a starting point.

Abubakar, I., et al. "Application of load monitoring in appliances’ energy management–A review." Renewable and Sustainable Energy Reviews 67 (2017): 235-245.

Ridi, Antonio, Christophe Gisler, and Jean Hennebert. "A survey on intrusive load monitoring for appliance recognition." 2014 22nd international conference on pattern recognition. IEEE, 2014.

Tekler, Z. D., Low, R., Zhou, Y., Yuen, C., Blessing, L., & Spanos, C. (2020). Near-real-time plug load identification using low-frequency power data in office spaces: Experiments and applications. Applied Energy, 275, 115391.

4)     Another direction is to use the same sensing technologies to collect information about the building’s state to automate the operation of different building systems to reduce energy usage. Some of the existing systems explored in the literature include lighting, HVAC, and plug loads, which are significant energy contributors. Please review the following papers as part of this review to provide a comprehensive overview.

Balaji, B., Xu, J., Nwokafor, A., Gupta, R. and Agarwal, Y., 2013, November. Sentinel: occupancy based HVAC actuation using existing WiFi infrastructure within commercial buildings. In Proceedings of the 11th ACM Conference on Embedded Networked Sensor Systems (pp. 1-14).

Tekler, Zeynep Duygu, et al. "Plug-Mate: An IoT-based occupancy-driven plug load management system in smart buildings." Building and Environment (2022): 109472.

Kandt, Alicen J., and Margarete R. Langner. Plug Load Management System Field Study. No. NREL/TP-7A40-72028. National Renewable Energy Lab.(NREL), Golden, CO (United States), 2019.

Zou, Han, et al. "WinLight: A WiFi-based occupancy-driven lighting control system for smart building." Energy and Buildings 158 (2018): 924-938.

5)     I highly suggest the authors to include a section to discuss about the future direction of these fields of study.

·       What are the current challenges?

·       What are the possible solutions to these challenges?

·       What are the potential innovations that these fields can expect?

6)     Some minor feedback to also take note off:

·       The captions for the figures are too brief and should clearly explain the figure without the readers referring to the text.

·       In the abstract, there is a double space in the sentence that starts with “ ..A detailed study.. “.

·       Again double space in the introduction sentence starts with “ … These issues..”.

Reviewer 2 Report

Dear Authors of the article "A Review : Agent in Home Energy Management and Smart Grid",

Please find below my recommendations for the improvement of your paper.

Before I made this review, I conducted an initial documentation process and I found that some paragraphs from your paper are similar to the article available at the address: https://doi.org/10.1109/ACCESS.2020.2981349.

For example:

- the last paragraph from page 1;

- an important part of the table 1;

- almost the entire chapter "3.1.2. Smart Meter Agents (SMA)"

- almost the entire chapter "4.1. Smart Home Agent concepts"

You should clarify this issue. Is the current manuscript proposal an extension or a remake of the previous published article?

Is the previous article yours? If yes, please clearly specify that the current manuscript is based on that article.

If not, please present the initial source.

The Introduction of your article is almost complete. Here, I recommend only to add a new paragraph where you present the research gap.

Maybe a formalization of the research hypotheses would be a good approach.

The title of sub-chapter "2.1. an overview on leveraging edge computing in smart grid" starts with small letter. Please correct.

Under the title of the "Figure 1. Edge Computing for Smart Grid.", please present the source. Is the figure generated from previous research? Or is your own vision?

The same remark for "Figure 2. Blockchain-Based Smart Grids."

Sub-chapter "3.1.3. Hybrid agents" has only 2 rows of text. Please avoid using such short "subchapters" in a scientific article.

You have 2 sub-chapters with the same number: 3.1.2. You must correct this situation.

The title of "4.3.9. Discussing" should be changed to "4.3.9. Discussions".

In the chapter "5. Conclusion and future works" I recommend you to also discuss about the limitations of your research and the managerial implications.

Regarding the managerial implications, I recommend you to add a new paragraph and present some economic aspects. Here you should cite the following resources: https://doi.org/10.3390/en14217396, https://doi.org/10.3390/jrfm14020048, https://www.researchgate.net/publication/325847421_a_profitability_regression_model_of_romanian_stock_exchange'S_energy_companies, https://doi.org/10.1016/j.renene.2016.10.029.

By including these references, you will have a strong argument for your research results.

Dear Authors,

I hope my recommendations will be useful for you to improve the manuscript during this round of review.

Kind Regards!

Reviewer 3 Report

The authors did a quite comprehensive analysis and literature survey on this topic and I am pleased with the thoroughness of their study. A few comments:

1. Surprisingly, the well-known and -adopted concept of Non-Intrusive Load Monitoring is not discussed. The fact that loads (appliances) can be detected accurately, data driven business cases can be built on top and also the fact that such a solution can run on the edge (devices, metering cabinets or smart meters) in my opinion should be mentioned. Two indicative example papers are: a) Athanasiadis, Christos, et al. "A scalable real-time non-intrusive load monitoring system for the estimation of household appliance power consumption." Energies 14.3 (2021): 767 and b) Athanasiadis, Christos L., Theofilos A. Papadopoulos, and Dimitrios I. Doukas. "Real-time non-intrusive load monitoring: A light-weight and scalable approach." Energy and Buildings 253 (2021): 111523.

2. A few acronyms are not properly defined in the manuscript. Please check that and edit accordingly the nomenclature in the beginning.

3. This is a review paper and it would be ideal to be as up-to-date as possible. There are only 3 papers from 2021 and none from this year in the references list. Please try to include more recent papers in the analysis also. 

Round 2

Reviewer 1 Report

Although some improvements made in the current manuscript, my concerns have not been fully addressed. I can see that the authors included more references to address many comments and unfortunately not all of them are relevant to the existing concern raised in the comments. On the other side, they have missed all suggested applications and references in my two previous comments below.

Please include these applications and suggested references to make this review complete.

3)     Under Section 4 Home Management systems, the authors should also focus on load monitoring technologies and their applications. Please review the following papers as a starting point.

Abubakar, I., et al. "Application of load monitoring in appliances’ energy management–A review." Renewable and Sustainable Energy Reviews 67 (2017): 235-245.

Ridi, Antonio, Christophe Gisler, and Jean Hennebert. "A survey on intrusive load monitoring for appliance recognition." 2014 22nd international conference on pattern recognition. IEEE, 2014.

Tekler, Z. D., Low, R., Zhou, Y., Yuen, C., Blessing, L., & Spanos, C. (2020). Near-real-time plug load identification using low-frequency power data in office spaces: Experiments and applications. Applied Energy, 275, 115391.

4)     Another direction is to use the same sensing technologies to collect information about the building’s state to automate the operation of different building systems to reduce energy usage. Some of the existing systems explored in the literature include lighting, HVAC, and plug loads, which are significant energy contributors. Please review the following papers as part of this review to provide a comprehensive overview.

Balaji, B., Xu, J., Nwokafor, A., Gupta, R. and Agarwal, Y., 2013, November. Sentinel: occupancy based HVAC actuation using existing WiFi infrastructure within commercial buildings. In Proceedings of the 11th ACM Conference on Embedded Networked Sensor Systems (pp. 1-14).

Tekler, Zeynep Duygu, et al. "Plug-Mate: An IoT-based occupancy-driven plug load management system in smart buildings." Building and Environment (2022): 109472.

Kandt, Alicen J., and Margarete R. Langner. Plug Load Management System Field Study. No. NREL/TP-7A40-72028. National Renewable Energy Lab.(NREL), Golden, CO (United States), 2019.

Zou, Han, et al. "WinLight: A WiFi-based occupancy-driven lighting control system for smart building." Energy and Buildings 158 (2018): 924-938.

Reviewer 2 Report

Dear Authors,

The new version of your article is improved and you addressed most of my recommendations. In order to finalize the shape of the article, you should add in the "Conclusion and future work" section a new paragraph containing the limitations of your research.

Best Regards!
